# The Molecular Landscape of Primary CNS Lymphomas (PCNSLs) in Children and Young Adults

**DOI:** 10.3390/cancers16091740

**Published:** 2024-04-29

**Authors:** Zhi-Feng Shi, Kay Ka-Wai Li, Anthony Pak-Yin Liu, Nellie Yuk-Fei Chung, Sze-Ching Wong, Hong Chen, Peter Yat-Ming Woo, Danny Tat-Ming Chan, Ying Mao, Ho-Keung Ng

**Affiliations:** 1Department of Neurosurgery, Huashan Hospital, Fudan University, Shanghai 200040, China; shizhifeng@fudan.edu.cn; 2Hong Kong and Shanghai Brain Consortium (HSBC), Hong Kong, China; 3Department of Anatomical and Cellular Pathology, The Chinese University of Hong Kong, Shatin, Hong Kong, China; kayli@cuhk.edu.hk (K.K.-W.L.); yf169chung@cuhk.edu.hk (N.Y.-F.C.); scwong@cuhk.edu.hk (S.-C.W.); 4Department of Paediatrics and Adolescent Medicine, The University of Hong Kong, Hong Kong, China; 5Department of Paediatrics and Adolescent Medicine, Hong Kong Children’s Hospital, Hong Kong, China; 6Department of Pathology, Huashan Hospital, Fudan University, Shanghai 200040, China; cherrychen30@126.com; 7Division of Neurosurgery, Department of Surgery, The Chinese University of Hong Kong, Shatin, Hong Kong, China; peterwoo@surgery.cuhk.edu.hk (P.Y.-M.W.); tmdanny@surgery.cuhk.edu.hk (D.T.-M.C.)

**Keywords:** CNS lymphoma, young adult, children

## Abstract

**Simple Summary:**

Primary CNS lymphomas (PCNSLs) in children and young adults are not common. In this study, we studied immunophenotype, gene rearrangement, homozygous deletion of CDKN2A and HLA, and mutation profiling of 34 PCNSL patients aged between 7 and 39 years and correlated the findings with clinical features and outcome. We found that the PCNSLs of the pediatric and young adult patients were immunophenotypically different from the PCNSLs of the older patients. They were also molecularly different from the latter group, as many of the common molecular findings identified in the latter were not present or common in the PCNSLs of the pediatric and young adult patients.

**Abstract:**

Pediatric brain tumors are often noted to be different from their adult counterparts in terms of molecular features. Primary CNS lymphomas (PCNSLs) are mostly found in elderly adults and are uncommon in children and teenagers. There has only been scanty information about the molecular features of PCNSLs at a young age. We examined PCNSLs in 34 young patients aged between 7 and 39 years for gene rearrangements of BCl2, BCL6, CCND1, IRF4, IGH, IGL, IGK, and MYC, homozygous deletions (HD) of CDKN2A, and HLA by FISH. Sequencing was performed using WES, panel target sequencing, or Sanger sequencing due to the small amount of available tissues. The median OS was 97.5 months and longer than that for older patients with PCNSLs. Overall, only 14 instances of gene rearrangement were found (5%), and patients with any gene rearrangement were significantly older (*p* = 0.029). CDKN2A HD was associated with a shorter OS (*p* < 0.001). Only 10/31 (32%) showed MYD88 mutations, which were not prognostically significant, and only three of them were L265P mutations. CARD11 mutations were found in 8/24 (33%) cases only. Immunophenotypically, the cases were predominantly GCB, in contrast to older adults (61%). In summary, we showed that molecular findings identified in the PCNSLs of the older patients were only sparingly present in pediatric and young adult patients.

## 1. Introduction

Primary CNS lymphoma (PCNSL) is usually a diffuse large B-cell lymphoma (DLBCL), with other types of lymphomas making up only a very small proportion of cases [1]. Late-middle-aged to elderly persons make up the bulk of the patients, and the median age for PCNSL is 66–67 years; it is very uncommon to find young patients with PCNSL [2,3]. PCNSL is even rarer in children. It has been estimated that there are only 15–20 new cases of pediatric PCNSLs in North America annually [4].

Immunohistochemically, PCNSLs most often show a non-germinal center B-cell-like (non-GCB) immunophenotype [5]. The activated B-cell-like (ABC) phenotype is believed to account for its poor prognosis [6]. Gene expression profiling shows that tumor cells are most closely related to late germinal center (exit) B cells [7]. Genomic alterations of PCNSLs have been well studied. Toll-like and B-cell receptor (TLR and BCR) signaling pathway aberrations have been identified in previous studies, and PCNSLs reveal a high frequency of mutations in genes such as MYD88, CARD11, and CD79B [8,9,10,11,12,13]. The L265P mutation of MYD88 has been shown to be especially prevalent in PCNSLs and suggested to be useful for diagnostic purposes [14,15]. NF-κB activation due to deregulation of TLR, BCR, and JAK-STAT signaling pathways is believed to lead to cellular proliferation in PCNSLs [5,14,16,17]. Homozygous deletion of CDKN2A, HLA Class II, and structural variants are also commonly found [13,18,19,20]. These molecular findings suggest PCNSL to be genetically similar to the “MCD”, “C5”, or “MYD88-like” subtypes of extracranial B-cell lymphomas, for which a derivation from long-lived memory B-cells has been suggested [21,22,23,24,25,26,27]. The outcome of PCNSL is poor compared to the bulk of extracranial DLBCLs [1]. The usual overall survival is only around three years [1]. And there is no specific target therapy for PCNSLs.

At present, PCNSLs are classified by the WHO Classification (2021) mostly as diffuse large B-cell lymphoma without further grading, unlike the common primary brain tumors within the WHO Classification [1]. Pediatric brain tumors have been noted to be different from their adult counterparts for many brain tumor entities in the WHO Classification of 2021, and brain tumors of the adult and young adolescent group are noted to be distinct as regards their clinical management, due to cognitive demands on this age group and psychological development during this period of personal growth. For gliomas, distinct molecular properties were found in the tumors of younger age groups [28,29]. However, there has only been scanty information about PCNSLs occurring in the pediatric age and young adult age groups.

Attarbaschi et al.’s study was a large retrospective series on PCNSLs in children and adolescents, but there were no molecular studies [30]. Similarly, another study on PCNSLs in the young contained no molecular information [31]. A very recent study on a small series of PCNSLs in children and young adults suggests that wild-type MYD88 defines a genetic group of PCNSLs with a favorable prognosis [2]. The number of cases was small (4 cases of mutant and 8 cases of wild-type MYD88 for this age group) in this series. Overall, the survival of pediatric and young patients with PCNSLs is shown to be much better than that of older patients in these studies. In this study, we postulated that the molecular characteristics of PCNSLs occurring in pediatric and young patients are different from those of older adults and investigated the molecular characteristics of 34 cases.

## 2. Materials and Methods

### 2.1. Patients

We went through the archives of the Prince of Wales Hospital, the Chinese University of Hong Kong, and Huashan Hospital, Fudan University, Shanghai, and identified 34 pediatric (age 18 or below) and young adults (age 19–40) with a diagnosis of PCNSL. While there is no absolute definition for the age of young adults, we used this age range as it has been used by others for similar studies in brain tumors [2,29,32]. All patients were immune-competent, except one patient who had a history of SLE. All cases were confirmed to have diseases confined to the CNS without systemic involvement.

### 2.2. Immunohistochemisty

All cases were histologically confirmed to be diffuse large B-cell lymphomas according to the WHO Classification (2021). Histological sections were reviewed, and all cases were immunohistochemically stained for CD20 (M0755; Dako, Glostrup Kommune, Denmark), CD10 (NCL-L-CD10-270, Novocastra, Newcastle upon Tyne, UK), BCL6 (NCL-L-Bcl-6-564, Novocastra), and MUM1/IRF4 (M7259; Dako). Immunohistochemical staining was performed on a Ventana BenchMark ULTRA Immunostainer. They were categorized according to Hans classification into germinal center B-like (GCB) or non-germinal center B-like (non-GCB) immunophenotypes [33]. EBER in situ hybridization (EBER-ISH) was carried out in all cases using commercial reagents and the Inform EBER probe (800–2842, Ventana Medical Systems, Oro Valley, AZ, USA).

### 2.3. Next-Generation Sequencing

As the mainstay treatment for PCNSLs is not surgical resection, the amount of tissue sampled was usually small. For next-generation sequencing (NGS), we tried to perform whole-exome sequencing (WES), but only in nine cases were we able to obtain sufficient DNA for the process. WES was performed in Shanghai, China (Sinotech Genomics Co., Ltd., Shanghai, China). The Agilent SureSelect Human All Exon V6r2 kit was used for target capture. Sequencing was performed on an Illumina HiSeq platform using a pair-end sequencing strategy.

### 2.4. Target Panel Sequencing

For cases without sufficient tissue for WES, we tried panel DNA sequencing using an established cancer panel (Appendix A); we were successful only in 15 of the remaining cases. So, altogether 24 cases were processed for either whole-exome sequencing or panel target sequencing.

Capture-based targeted sequencing with a custom-designed 415-gene panel was performed on the DNA from the FFPE samples. DNA was purified using truXTRAC FFPE kits (Covaris, Woburn, MA, USA) according to the manufacturer’s protocol. Libraries were prepared using the KAPA HyperPrep kit (Roche, Basel, Switzerland) and enriched with a custom-designed 415-gene solution-based hybrid capture panel (IDT, Coralville, IA, USA). This cancer-related gene panel targeted approximately 1.3 Mb of the human genome. The gene list is shown in Appendix A. The libraries were sequenced on the Illumina HiSeq platform at 150 bp paired-end to achieve a mean coverage of ≥200×. The data analysis was performed as described in [34,35].

### 2.5. Sanger Sequencing

We studied the remaining ten cases, which failed both WES and target panel sequencing, using Sanger sequencing for the hotspot MYD88 mutation L265P, and we were successful in seven cases. In brief, DNA was extracted from FFPE tissues using a KAPA Express Extract kit (Roche) upon microdissection. PCR amplification was performed using KAPA 2G Fast Readymix (Roche) and the primers flanking the L265 mutation hotspot of the MYD88 gene (forward: 5′-CCCACCATGGGGCAAGG-3′ and reverse: 5′-GGTGTAGTCGCAGACAGTGATGAA-3′). The PCR product was purified and sequenced using a BigDye v1.1 kit (ThermoFisher, Waltham, MA, USA).

### 2.6. FISH Analysis

Detection of gene fusions for BCL2, BCL6, CCND1, IRF4, IGH, IGL, IGK, and MYC, CDKN2A homozygous deletion (HD), and HLA HD was achieved by FISH with the use of commercial probes. Details of the probes can be found in Appendix A. In short, 4 μm thick FFPE sections were deparaffinized in xylene, treated with 1 M sodium thiocyanate, digested in pepsin solution, rinsed in milli-Q water, and dehydrated. The labeled probes were denatured and hybridized to the sections overnight. The sections were then washed, stained with Vectashield mounting medium, and visualized under a Zeiss Axioplan fluorescence microscope. The samples were considered positive when a break-apart signal was noticed in >15% of the evaluated nuclei [36]. The samples were considered positive for HD when >20% of tumor cells showed loss of two signals, as previously used by us [37].

## 3. Results

### 3.1. Clinical Characteristics of This Study Cohort

This series of pediatric and young adult PCNSLs comprised 34 patients, and their age range was from 7 to 39. The mean and median ages of the cohort were 27.2 and 30 years, respectively. Eight patients were pediatric, and their age ranged from 7 to 16 years (Figure 1). Interestingly, 12/34 (35%) cases were mid-line.

Survival information was available in 33 cases. However, the survival of one patient was excluded from the analyses as the patient passed away very soon after the operation due to post-operative complications. The mean and median overall survival was 78.4 ± 7.6 months and 97.5 ± 33.0 months, respectively. Details of post-operative treatment regimens were unavailable for four patients. All except three patients received high-dose methotrexate as primary treatment. One patient died too soon to receive chemotherapy, and two patients received chemotherapy regimens other than high-dose methotrexate. Six patients received rituximab as an adjuvant treatment.

There was no significant difference in OS between pediatric patients (aged 18 or below) and young adults (*p* = 0.974; Appendix A). It has been suggested by Guney et al. [2] that PCNSLs below the age of 25 constitute a different molecular group, and those between 25 and 40 have molecular characteristics similar to those of older patients, and wild-type MYD88 tumors occurring in younger patients may have a better prognosis. In our cohort, there was no difference in survival between patients aged >25 years and those aged ≤25 years (*p* = 0.485).

### 3.2. Immunophenotype

Immunophenotyping based on Hans classification for B-cell lymphomas was available for 33 patients, and there was insufficient tissue for further immunohistochemical staining for one patient. Twenty (61%) cases were classified as GCB, and the rest as non-GCB. Representative cases of the former are shown in Figure 2A–C. Kaplan–Meir analysis revealed that non-GCB tumors had a trend for a shorter overall survival (*p* = 0.070). It has been well documented that the non-GCB type accounts for the majority of PCNSLs in adults, and GCB tumors are associated with favorable survival [38,39]. In extracranial lymphomas in children, the GCB subtype is predominant, and our finding is in concordance [40]. EBER in situ hybridization was only positive in one case, and the patient had SLE as a background disease.

### 3.3. Breakapart FISH

All cases were successfully tested for common breakapart molecular events for B-cell lymphomas and PCNLs, BCL6, BCL2, CCND1, IGH, IGL, IGK, and MYC. These tests are often used for the diagnosis of extracranial lymphomas. Out of a total of 272 tests, only 14 instances for gene rearrangement (5%) were found (Figure 3): six for BCL6, three for CCND1, two for IGH, and one for MYC, IRF4, and IGL each. One tumor (case 19) in a 31-year-old female showed a double-hit genotype, which is defined by two chromosome translocations involving MYC and BCL2 and/or BCL6 rearrangements. The double-hit genotype is usually associated with aggressive behavior [41,42], and this patient had an overall survival of 28.8 months. No triple-hit genotype was found in this cohort. Patients with BCL6 rearrangements were significantly associated with older age (*p* = 0.034; 4.5 ± 3.7 vs. 25.6 ± 9.5 years old). In fact, patients carrying any rearranged genes in this study were older than those with none (*p* = 0.029; 31.9 ± 6.6 vs. 24.6 ± 9.8 years old). Rearranged genes were not associated with other clinical factors, including sex, tumor location, and immunophenotypes. In contrast to the PCNSLs in older patients in whom BCL6 rearrangement is associated with an inferior survival [43,44], there was no prognostic impact for BCL6 rearrangement in our cohort. Furthermore, overall survival was not different between patients with MYC, BCL2, or BCL6 fusions and those without. Appendix A lists the frequencies and prognostic significance of the molecular alterations studied in this study.

### 3.4. CDKN2A Homozygous Deletion

CDKN2A inactivation has been well documented in PCNSLs [13,18,45]. Braggio et al. demonstrated CDKN2A deletion or mutation in over 80% of the PCNSLs of older patients [17]. Using FISH, homozygous deletion of CDKN2A was seen in 5 cases (15%) in this cohort and was found in both pediatric and young adults, in both hemispheric and mid-line cases and in both GCB and non-GCB cases (Figure 4A,B). Target sequencing also revealed that one case had a nonsense mutation of CDKN2A. CDKN2A deletion and CDKN2A deletion/mutation were associated with poorer survival (*p* < 0.01 and *p* = 0.001, respectively) (Figure 4C,D). However, in this study, there were only 5 cases with CDKN2A deletion, so the number was small, and cautious interpretation is recommended. However, CDKN2A homozygous deletion is a well-known prognosticator in PCNSLs. Unlike the situation with BCL6 gene rearrangement, CDKN2A deletion was not associated with age (Appendix A).

### 3.5. HLA Homozygous Deletion

Previous studies showed loss of HLA locus in ~60% of PCNSL in older patients [46], and HLA loss has been associated with an inferior outcome. Thus, we investigated HLA copy numbers in our cohort using FISH. We found HLA homozygous deletion in 8/34 (23%) tumors. HLA loss was not associated with age, sex, tumor location, or immunophenotype. There was no prognostic association with HLA loss in this study (Appendix A).

### 3.6. Mutation Profiling

We performed whole-genome sequencing and capture-based target sequencing on PCNSL samples where there was sufficient material and an adequate quality of nucleic acid. For cases that failed the quality check for NGS, we performed MYD88 Sanger sequencing for the L265P mutation. Overall, we did not find any difference in mutation burden between pediatric patients and young adults (*p* = 0.237; 4.7 ± 6.7 vs. 8.2 ± 6.3) for cases that were studied using NGS. Mutation burden was not associated with sex, location, or immunophenotype. The commonest mutations found were MYD88 (*n* = 10), GNA13 (*n* = 9), KMT2C (*n* = 9), NFKB1E (*n* = 8), CARD11 (*n* = 8), and TP53 (*n* = 6) (Figure 1 and Appendix A).

The NF-κB pathway is associated with cell division and includes the genes PIM1, MYD88, CD79B, IRF4, GNA13, NFKBIE, CARD11, and TNFAIP3. It has been found to be altered in up to 80–90% of the PCNSLs of older patients [13,47]. We found that the frequency of mutations of NF-κB pathway-related genes was lower (64%, 16/25) (Figure 1). Furthermore, NF-κB pathway alterations were associated with older age (*p* = 0.021). Only 29% of pediatric patients aged below 18 years showed alterations in NF-κB pathway; in contrast, 78% of young adult PCNSLs showed alteration. However, an individual gene of the NF-κB pathway was not associated with age.

Thirty-one tumors were successfully tested for MYD88 mutations. In 9 cases, the results were derived from WES. In 15 cases, they were obtained from panel sequencing. In 7 cases, there were insufficient tissues for either study, and only Sanger sequencing for hotspot MYD88 L265P mutation was performed. This was the best we could do given the small amount of tissues available in some cases, as is the usual case for biopsies for PCNSLs. Overall, ten MYD88 mutations were identified, and of these, only three were L265P mutations. Nine of the ten mutations occurred in the young adult patient group. All L265P mutations were found in older patients aged above 25 years. Five cases carried MYD88 mutations that affected the amino acid M232T, and three of these were from patients aged 20 years or below. This mutation in MYD88 has previously been reported very rarely in the PCNSLs of older patients [17,48]. The two remaining MYD88 mutations were not well known in PCNSLs. All patients with MYD88 mutations in this study were alive. There was no survival difference between patients with MYD88 mutations and those without (*p* = 0.543). Given the imperfect situation due to the variability of methods used in testing for MYD88 mutation arising from the small quantity of materials, we compared cases shown to have MYD88 mutations using NGS with those that were negative for mutations using NGS, and there was also no survival difference (*p* = 0.523).

The CARD11 gene is a positive regulator of the NF-κB pathway and was mutated in 8/24 (33%) tumors, and all mutations were found in patients aged above 18 years. GNA13, an α subunit of a heterotrimeric G protein [49], was the commonest gene other than MYD88 to be mutated in our cohort, and it was found to be mutated in 9/24 (38%) cases successfully processed for NGS. Interestingly, seven such cases, including pediatric and young adult tumors, also showed MYD88 mutation, and GNA13 mutations were highly correlated with MYD88 mutations (*p* = 0.002). Neither CARD11 nor GNA13 mutations were associated with a survival impact.

Recently, PCNSL has been related to the so-called ‘MCD’ subgroup [3], which is a genetic subset of lymphomas with gain-of-function mutations in both MYD88 L265P and CD79B [23]. In our cohort, only one adult tumor showed both MYD88 and CD79B mutations. In fact, mutations of CD79B, which was shown to lead to activation of BCR signaling and subsequently the translocation of the NF-κB protein [50,51], were found in two cases in this study. The gene coding for the PIM1 serine/threonine kinase, which stabilizes the main subunit of NF-κB, RelA/p65, has been shown to be mutated in 70–80% of the PCNSLs of older patients [14,52,53,54]; it was mutated in only one young adult case in this cohort.

Other than the NF-κB pathway, we found high frequencies of mutations in the epigenetic pathway (75%). In contrast to NF-κB pathway, alterations in the epigenetic pathway were not associated with age groups in this study. KMT2C and KMT2D mutations were the commonest mutated genes in this pathway and were found in 38% and 21% of cases studied successfully (Figure 1). KMT2D was associated with a mid-line location (*p* = 0.027). No other association was detected.

Other mutations identified in PCNSLs include regulators of immune function and the cell cycle (Figure 1). PRDM1, PRDM15, JAK1, JAK2, IRF4, and IRF8 participate in immune function and signaling. Mutations in these genes were detected in 38% (9/24) samples. Interestingly, all mutations were detected in adult patients aged above 18 years (*p* = 0.008). Moreover, mutations were more prevalent in males (*p* = 0.019). TP53, RB1, BTG1, CCND3, and CDKN2A belong to the cell cycle regulation pathway. Mutations in this pathway were detected in 42% (10/24) samples. TP53 was the most frequently mutated gene in this pathway (25%, 6/24), followed by CCND3 (17%, 4/24) and BTG1 (8%, 2/24). Four p53 mutated cases were concurrently mutated for MYD88. Mutations in this pathway were not associated with age, sex, or location. A summary of the significant findings of this study can be found in Table 1.

## 4. Discussion

Previous series of PCNSLs in pediatric and young patients have reported a generally favorable prognosis [2,30,55,56] compared to the survival of about 3 years usually referenced for older patients with PCNSLs [1]. The median overall survival of 78.4 months in this series of pediatric and young adult PCNSLs is consistent with the literature, but there was no difference in OS between pediatric and young adult patients in our cohort. However, many patients in this cohort were post-operatively managed by different teams outside our hospitals, and there were differences in treatment regimens; the cases were also retrieved from archives over a number of years due to its rarity.

For comparison of our findings to previous studies, Appendix A lists the frequencies of molecular changes detected in the older patients with PCNSLs as per the literature. The immunophenotypes of PCNSLs in older patients have been shown to be predominantly non-GCB in previous studies, including both immunohistochemical studies and nanostring expression studies [5,11]. In one study, three quarters of the PCNSLs in older patients were non-GCB [57]. However, in our cohort of young patients, only 39% of the cases were of non-GCB and the majority of the tumors were of the GCB phenotype (61%). Furthermore, in the PCNSLs of older adult patients, GCB immunophenotype has been shown to be associated with a favorable outcome [57], but we could only find a trend towards a favorable outcome, probably due to the small size of the cohort.

MYD88, which induces activation of the NF-κB signaling pathway, has been shown to be an important mutation found in PCNSLs in older patients [13,18,53], and it is part of the “MCD” genetic subgroup of extracranial diffuse large B-cell lymphoma (DLBCL); it is defined as the co-occurrence of MYD L265P and CD79B mutations [58]. MYD88 L265P is often found in lymphoplasmacytic lymphomas occurring outside the brain. In Radke et al.’s paper on PCNSLs of older patients, all of the MYD88 mutations were L265P [53]. In Nayyar et al.’s study [14], 42 of 63 (67%) PCNSLs showed MYD88 L265P [14]. Guney et al. regarded MYD88 hotspot mutations as characteristic of the PCNSLs of older patients [2]. It has even been suggested that screening of L265P mutations might be used in diagnostic pathology [15]. However, it should be noted that a significant number of MYD88 mutations found in PCNSLs by Zhu et al. were of the non-L265P type [59].

There was only one recent study on the molecular characteristics of PCNSLs in pediatric and young patients [2]. The authors categorized two groups of such PCNSLs based on their MYD88 genotypes, with the wild-type MYD88 cases having a better survival. There were only four mutant cases and eight wild-type cases in the young patients’ group (<25 years old) in this paper [2]. We did not find MYD88 mutations to have prognostic significance in our cohort. All young adult PCNSLs in our cohort with tumors carrying MYD88 survived. No prognostic impact has been reported for the presence of MYD88 mutations in different series. In older adults, MYD88 mutations were not associated with survival by Gonzalez-Aguilar et al. [18]. Similarly, Sethi et al. showed that MYD88 mutation or protein expression did not predict both PFS and OS in the PCNSLs of older patients [60].

In our cohort, 4/8 cases with p53 mutations also showed MYD88 mutations. Overall, 17% of our cohort exhibited co-occurrence of MYD88 and P53 mutations. A previous study indicated that 14% of extracranial diffuse large B-cell lymphoma carried both p53 and MYD88 mutations [61], but this has not been reported in pediatric PCNSLs. Similarly, 7/9 cases with GNA13 mutations also showed MYD88 mutations. P53 mutations were only rarely found in the PCNSLs of older patients from previous studies [62,63]. KMTD2 and PIM1 mutations were the other frequently mutated genes found in PCNSLs, according to one study [11]. However, KMTD2 mutations were found in 5/24 cases and PIM1 mutations were found in one patient in our study.

IGH, IGK, and IGL breakapart tests are often used to investigate B-cell lymphomas diagnostically and have also been studied in PCNSLs of older patients [64]. These fusion events were detected in 16/30 cases of PCNSLs in one study [53]. Similar findings of IGH in the PCNSLs of older patients were reported by Montesinos-Rongen [64]. However, these events were rare in this series of PCNSLs in pediatric and young patients. Chapuy et al. showed that BCL6 fusion was the commonest chromosomal rearrangement in PCNSLs of older patients [13]. Cady et al. showed that translocation events involving BCL6, IGH, and MYC and del(6)(q22) were often found in PCNSLs [44]. Other sites of IGH fusion were also identified in PCNSLs [36]. Radke et al. detected recurrent IGH-BCL6 fusions in PCNSLs [53]. However, BCL6 breakapart events were only found in 6/34 patients in our cohort of pediatric and young adult patients, and none of them occurred concurrently with IGH rearrangements. Cases of IGH-BCL2 fusions in the PCNSLs of older patients were also found by Radke et al., but our BCL2 FISH was negative for the entire cohort [53]. Triple hits of MYC, BCL2, and BCL6 breakaparts as reported in the PCNSLs of older patients were not seen in this study [42]. No BCL2 or MYC rearrangements were found in PCNSLs in young patients by Guney et al. [2].

CDKN2A homozygous deletion is a poor prognostic biomarker in B-cell lymphomas in general and has also been reported to be a common molecular abnormality in the PCNSLs of older patients [53,65]. Even though we found the poor prognostic significance of CDKN2A homozygous deletion, a finding similar to other studies in older patients, we should be cautious in the interpretation as our cohort was of children and young adults and only a small number of cases in this cohort showed alteration.

## 5. Conclusions

In summary, we showed that the PCNSLs of pediatric and young adult patients were immunophenotypically different from the PCNSLs of older patients. They were also molecularly different from the latter group, as many of the common molecular findings identified in the latter were not present or common in the PCNSLs of pediatric and young adult patients.

## Figures and Tables

**Figure 1 cancers-16-01740-f001:**
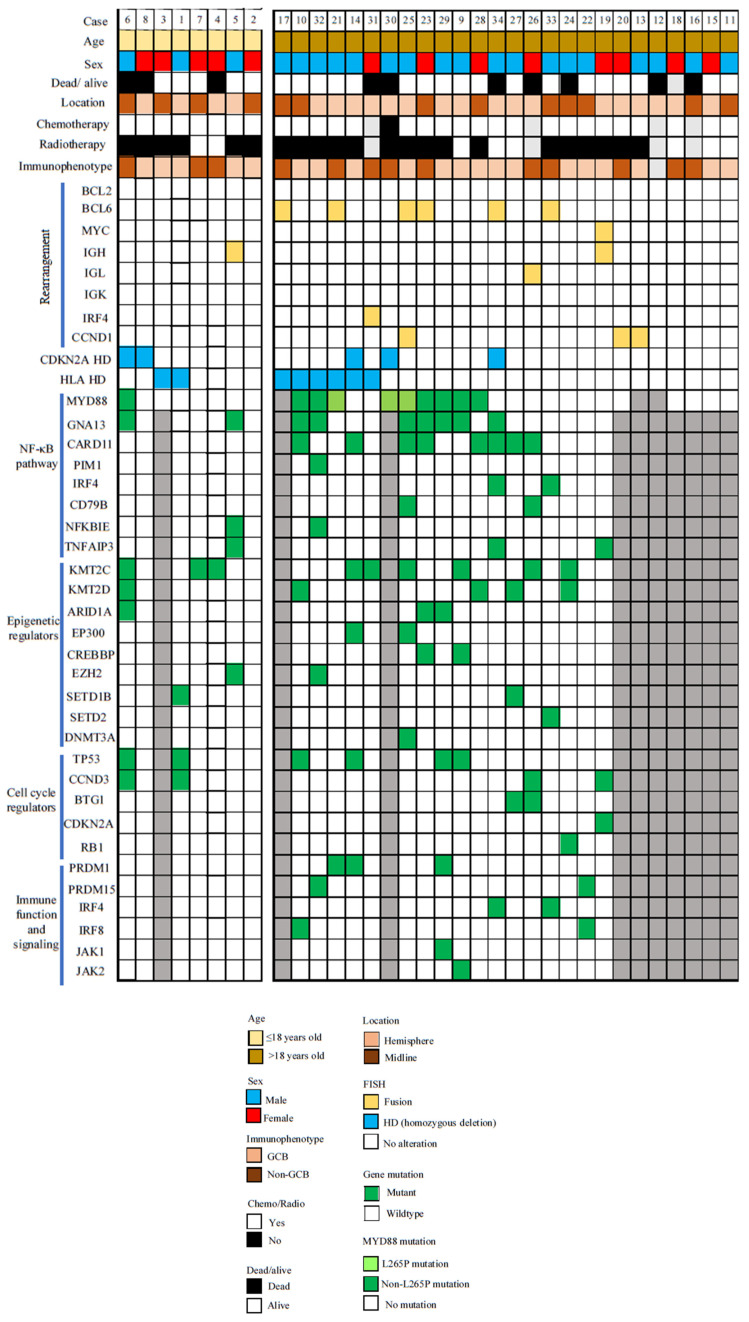
Oncoprint showing the clinical and molecular data of pediatric and young adult PCNSLs.

**Figure 2 cancers-16-01740-f002:**
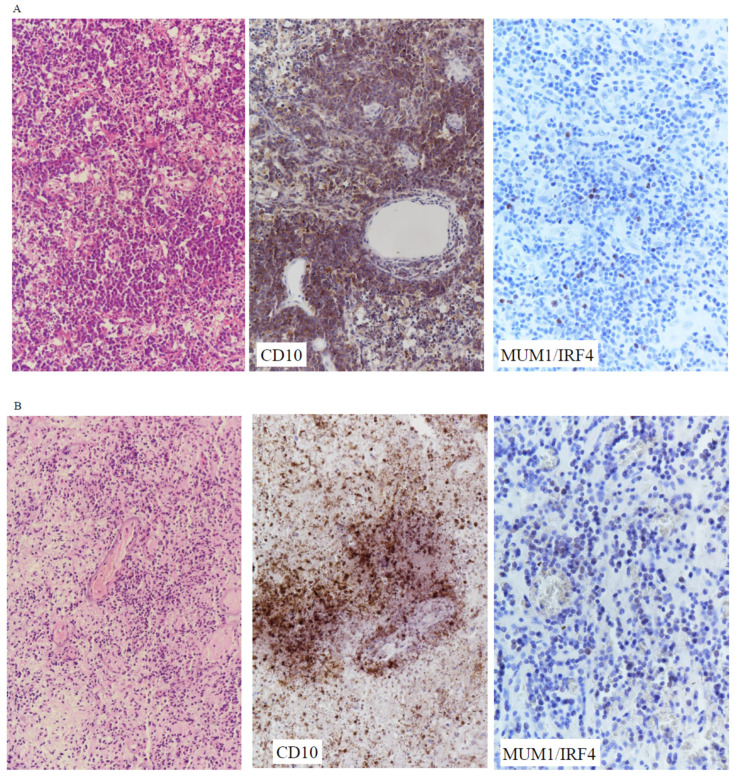
Pediatric and young adult PCNSLs with GCB phenotype. (**A**) 34/M, frontal lobe tumor (case 27). The patient had an overall survival of 83.5 months (alive). (**Left**) H&E (×200). (**Middle**) CD10 was positive (×200). (**Right**) MUM1 was negative (×200). (**B**) 33/M, cerebellar tumor (case 22). Overall survival was 37.8 months (alive). (**Left**) H&E (×200). (**Middle**) CD10 was positive (×200). (**Right**) MUM1 was negative (×200). (**C**) 11/F, T1-5 spinal cord tumor (case 2). Overall survival was 105 months (alive). (**Left**) H&E (×200). (**Middle**) CD10 was positive (×200). (**Right**) MUM1 was negative (×200).

**Figure 3 cancers-16-01740-f003:**
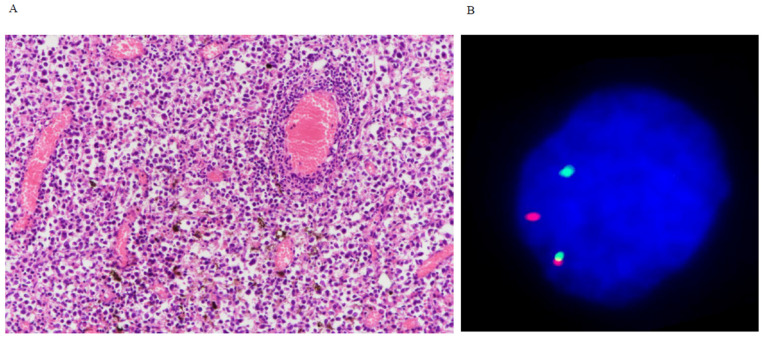
(**A**) H&E of a tumor from a 33-year-old female (case 23) diagnosed with PCNSL (×200). (**B**) FISH revealed BCL6 gene rearrangement. The split red and green signals represent a rearranged allele, and the fused signal represents a non-rearranged allele.

**Figure 4 cancers-16-01740-f004:**
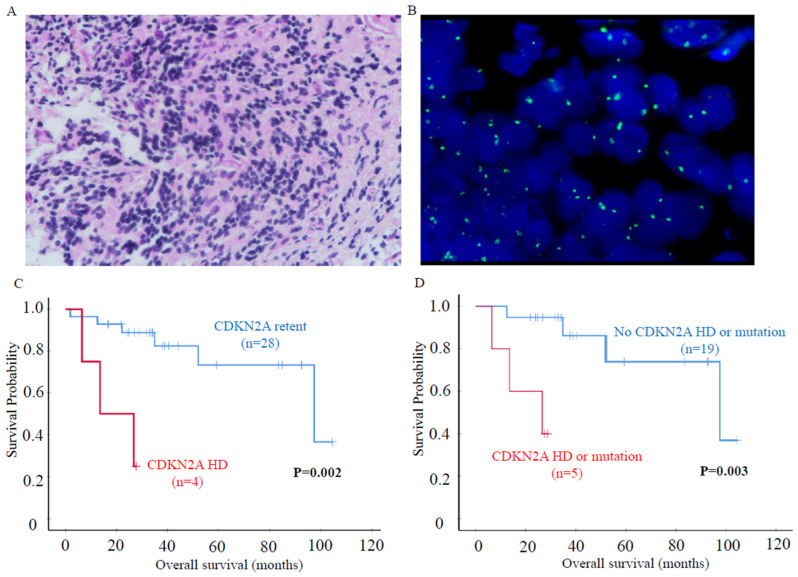
(**A**) H&E of a tumor from a 16-year-old boy (case 6) with a PCNSL at the brain stem (×400). (**B**) FISH revealed a CDKN2A homozygous deletion (HD). Tumor cells showed no CDKN2A orange signals, and only green reference signals were detected (Vysis). (**C**) CDKN2A HD and (**D**) CDKN2A HD or mutation were associated with poor survival.

**Table 1 cancers-16-01740-t001:** A summary of the significant findings of this study.

Primary CNS lymphomas (PCNSLs) of children and young adults have only very rarely been studied for molecular landscape.
34 such cases now studied by targeted sequencing, IHC and FISH.
Majority of cases (61%) were immunohistochemically GCB subtype.
Only rare instances (5%) of gene rearrangement was detected by FISH.
Only 10 cases (32%) showed MYD88 mutations and only three mutations were of L265P.
In summary, common molecular findings in PCNSLs of older patients were only sparingly present in pediatric and young adult cases.

## Data Availability

Data available on request from the authors.

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
