# Peer review of "The Molecular Landscape of Primary CNS Lymphomas (PCNSLs) in Children and Young Adults"

_cancers, 2024, doi:10.3390/cancers16091740_

Round 1
Reviewer 1 Report
Comments and Suggestions for Authors
The authors analyzed the immunophenotypic and molecular genetic differences of pediatric and young adult patients with primary CNS lymphomas versus older patients.
Specific Points of Criticism:
(1) Lines 110 + 192: "Hans classification" (or “Hans´ classification”), but not "Han´s classification".
(2) Figure 4: These are very low numbers on the CDKN2A HD-arm which clearly prevent a significant statistical analysis.
(3) Percentages: Is it relevant and useful to present the percentages with digits after the period (12.5 %) instead of only full numbers (12%) when such few cases are involved? Suffice to use the whole numbers.
(4) Summary table: A summary with the most significant findings would be useful, for example with bullet points and key words (instead of full sentences).
Author Response
(1) Lines 110 + 192: "Hans classification" (or “Hans´ classification”), but not "Han´s classification".
Response:
Thank you for your suggestive recommendation. We amended to “Hans classification.” The changes have been highlighted.
(2) Figure 4: These are very low numbers on the CDKN2A HD-arm which clearly prevent a significant statistical analysis.
Response:
Thank you for the criticism. We agree that the numbers were low. We made the precautionary statements concerning statistics in the Results as well as in the Discussion. Nonetheless, as CDKN2A HD is a well-known poor prognostic marker in lymphomas in general as well as in PCNSLs, we feel that we cannot avoid discussing it. In Results, we now state “However, in this study, there were only 5 cases with CDKN2A deletion, so the number was small and cautious interpretation is recommended. However, CDKN2A homozygous deletion is a well-known prognosticator in PCNSLs.” In Discussion, we now state “Even though we found the poor prognostic significance of CDKN2A homozygous deletion, a similar finding to other studies in the older patients, we should be cautious in the interpretation as our cohort was of children and young adults and only a small number of cases in this cohort showed the alteration.”
(3) Percentages: Is it relevant and useful to present the percentages with digits after the period (12.5 %) instead of only full numbers (12%) when such few cases are involved? Suffice to use the whole numbers.
Response:
Thank you for your constructive suggestion. We now round off all percentages to whole numbers. The changes have been highlighted.
(4) Summary table: A summary with the most significant findings would be useful, for example with bullet points and key words (instead of full sentences).
Response:
We now include Table 1 listing the most significant findings of this study.
Reviewer 2 Report
Comments and Suggestions for Authors
1. Have the prognostic significance of other genes besides CDKN2A HD and CDKN2A HD or mutation been tested? Why is this the only one described? as the only one based on literary data or based on verification by the authors?
2. I would like to see a table comparing the frequency of occurrence of certain mutations according to the authors’ data and according to data for an older age group in order to understand the differences.
3. Is there currently potential for using targeted therapy to identify certain mutations? If yes, please briefly describe the therapeutic implications of your results.
Author Response
1. Have the prognostic significance of other genes besides CDKN2A HD and CDKN2A HD or mutation been tested? Why is this the only one described? as the only one based on literary data or based on verification by the authors?
Response:
The prognostic significance of all of the positive genetic findings (mutations and CNVs) were actually tested, not only CDKN2A HD/mutations. We concede because of the number of genes and CNVs, sometimes the message has not been clear enough. We now added Supplementary Table 4 listing the prognostic significance of all genes listed in the oncoprint.
2. I would like to see a table comparing the frequency of occurrence of certain mutations according to the authors’ data and according to data for an older age group in order to understand the differences.
Response:
In Supplementary Table 5, we have now summarized the results of similar tests done in the key papers in PCNSLs in the older patients. A strict side-by-side comparison with our results, due to differences in methodology, would be misleading in our view and it is not possible to list all the tests.
3. Is there currently potential for using targeted therapy to identify certain mutations? If yes, please briefly describe the therapeutic implications of your results.
Response:
As far as we know, there is no specific target therapy for PCNSL as a distinct entity. We made this point too in the Introduction (page 5, line 21) and it is highlighted.